# Size-dependent diffusion controls natural aging in aluminium alloys

Phillip Dumitraschkewitz[1]*, Peter J. Uggowitzer [1,2], Stephan S.A. Gerstl[2,3], Jörg F. Löffler [2] & Stefan Pogatscher [1]*

A key question in materials science is how fast properties evolve, which relates to the kinetics of phase transformations. In metals, kinetics is primarily connected to diffusion, which for substitutional elements is enabled via mobile atomic-lattice vacancies. In fact, non-equilibrium vacancies are often required for structural changes. Rapid quenching of various important alloys, such as Al- or Mg-alloys, results for example in natural aging, i.e. slight movements of solute atoms in the material, which significantly alter the material properties. In this study we demonstrate a size effect of natural aging in an AlMgSi alloy via atom probe tomography with near-atomic image resolution. We show that non-equilibrium vacancy diffusional processes are generally stopped when the sample size reaches the nanometer scale. This precludes clustering and natural aging in samples below a certain size and has implications towards the study of non-equilibrium diffusion and microstructural changes via microscopy techniques.

[1] Chair of Nonferrous Metallurgy, Department of Metallurgy, Montanuniversitaet Leoben, Franz-Josef-Str. 18, 8700 Leoben, Austria. [2] Laboratory of Metal Physics and Technology, Department of Materials, ETH Zurich, 8093 Zurich, Switzerland. [3] Scientific Center for Optical and Electron Microscopy, ETH Zurich, 8093 Zurich, Switzerland. *email: phillip.dumitraschkewitz@unileoben.ac.at; stefan.pogatscher@unileoben.ac.at

The kinetics of phase transformations is a central topic in materials science. Frequently, nonequilibrium vacancies, which are induced by rapid cooling, irradiation, sputtering, or plastic deformation[1–4], are required to activate structural changes. Already in 1911 Wilm[5] discovered hardening of Al alloys during room temperature storage when trying to increase the hardness of an Al alloy in a procedure similar to steel quenching. This effect was given the name natural aging (NA). The hardness increase during room temperature storage is attributed to the formation of nanometer-sized unordered accumulations of solute atoms in the material, so-called clusters. The kinetics of NA strongly depends on nonequilibrium vacancies[2], and the effect has great importance for all classes of novel high-strength Al alloys[6]. Lately, it has also attracted progressing interest for magnesium alloys[7–9], due to the improvement of characterization methods, i.e., microscopy and microanalysis techniques with atomic (transmission electron microscopy, TEM) and near-atomic (atom probe tomography, APT) resolution[10–13]. An Al alloy in which NA has been studied intensively over the past 20 years[14] is of type AlMgSi. Here, NA has a detrimental effect on the final mechanical properties[15] and this limits the extended use of various AlMgSi alloys in lightweight applications[16]. However, it should be also noted that NA can be beneficial for gaining strength in other low-alloyed AlMgSi alloys[17], Mg alloys[18], or AlCuMg alloys[6]. In Fig. 1, we illustrate the complex effect of NA in an AlMgSi alloy (EN AW 6016) via differential scanning calorimetry (DSC). Even for short times after quenching, the AlMgSi alloy shows a pronounced change in DSC traces. With increased NA time enhanced cluster formation results in increasing endothermic cluster dissolution upon heating. The formation of the main hardening phase ($\beta''$) is retarded[19], indicating the negative effect of NA for this AlMgSi alloy. Note that clustering itself also increases hardness upon NA (see Supplementary Note 1 and Supplementary Fig. 1).

Since the emergence of APT many studies involving the direct observation of clusters in Al alloys have been documented[14,20–25]. However, they have generated contradictory results with regard to the sequence of cluster formation, especially concerning the early stages of clustering[6,14,21–24]. In the following we illustrate that the disagreement may be caused by an incorrect assessment of the NA time. In fact, the NA time is generally determined by the total time which the samples or components experience at room temperature after quenching, $\Delta t_{NA}$. An explicit distinction between the NA of finished APT samples or bulk material is, however, lacking in many cases. Here, we demonstrate that the above definition of $\Delta t_{NA}$ is problematic because only the time over which the material is exposed at bulk dimensions governs the clustering amount. This is expected to be a universal effect and not limited to the alloy of this study. We illustrate the problematic definition of $\Delta t_{NA}$ by means of two differently designed experiments. The first uses "nano aging" where the NA is performed in situ in the atom probe on finished needle-shaped specimens. In the second experiment we perform "bulk aging" where the NA time at bulk dimensions is varied, but the total NA time (the sum of "bulk aging" and "nano aging") is kept constant. Nano aged specimens do not show an increasing signal for clustering over NA time, whereas bulk aged samples do. From thermo-kinetic calculations it is concluded that nonequilibrium vacancies are annihilated at the surface of the nano-sized specimen, effectively stopping NA. Since substitutional diffusion is mainly vacancy mediated, the obtained effect can be considered universal for nonequilibrium substitutional diffusion in metals.

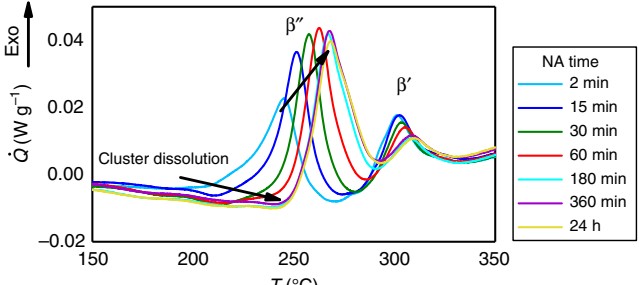

**Fig. 1** Excess heat flow (DSC heating curves) at varying NA times after quenching of an AlMgSi alloy. Increasing NA time progressively increases cluster dissolution and delays the formation of the main hardening precipitate ($\beta''$) upon heating

## Results

**Nano aging.** For nano aging the sample preparation and transfer took place under arctic conditions, and a special atom probe equipped with a novel cryo-transfer system[12,26,27] was used to suppress any diffusion during preparation and manipulation (see Methods section for details). Figure 2a provides an overview of the in situ NA sample nano_aged_01. We use the ratio of the cumulative sums of the radial distribution function for the interactions of species A and B (AB) for spatial analysis (Eq. (1)[26]), where values >1 indicate clustering.

$$f_{AB}(r) = \frac{\sum_{R=0}^{r} \text{RDF}_{AB}(R)}{\sum_{R=0}^{r} \text{RDF}_{AB,\text{random}}(R)}. \quad (1)$$

This formalism has the advantage of being parameter-free[28], in comparison to a cluster-finding algorithm which strongly depends on the input parameters (see also Supplementary Note 4), and still comprises the information from the whole spatial distribution of chosen solutes within a given radius $r$. A more detailed description of the analysis method can be found in the Supplementary Note 2. Spatial analyses of the solute species Mg and Si, shown in Fig. 2b, reveal no significant increase for the Mg–Mg, Si–Mg, Mg–Si, or Si–Si interactions due to in situ aging (see also Supplementary Fig. 2). The solute distribution only differs slightly from a random comparator and does not increase over the applied NA time, $\Delta t_{NA}$, up to 3 weeks. A trend that may be visible is a small decrease in Si–Si interactions with in situ NA after the "10 min" NA measurement (Fig. 2b), which is contrary to the expected trend for clustering. This may be attributed to the method specific field-evaporation artifact of Si rather than to an increased NA time (see Supplementary Note 3). The observation that no clustering occures in nano-sized specimens is contrary to all expectations and literature results on NA, also to those from DSC in Fig. 1, where microstructural changes were already obvious after several minutes of $\Delta t_{NA}$. An explanation for this unexpected result follows.

**Vacancy annihilation.** Because clustering upon NA is in general a substitutional diffusion process at room temperature, it only happens due to the availability of nonequilibrium vacancies from quenching[2]. In general, diffusion of a pure element by a random walk of vacancies can be described by

$$D = \frac{1}{6}a^2 n c_v \omega, \quad (2)$$

where $D$ is the self-diffusion coefficient, $a$ denotes the jumping distance, $n$ is the coordination number, $c_v$ is the vacancy concentration, and $\omega$ is the jump frequency of the vacancy[29]. $c_v$ directly influences the self-diffusion coefficient. While the description of solute diffusion in a matrix is more complex and

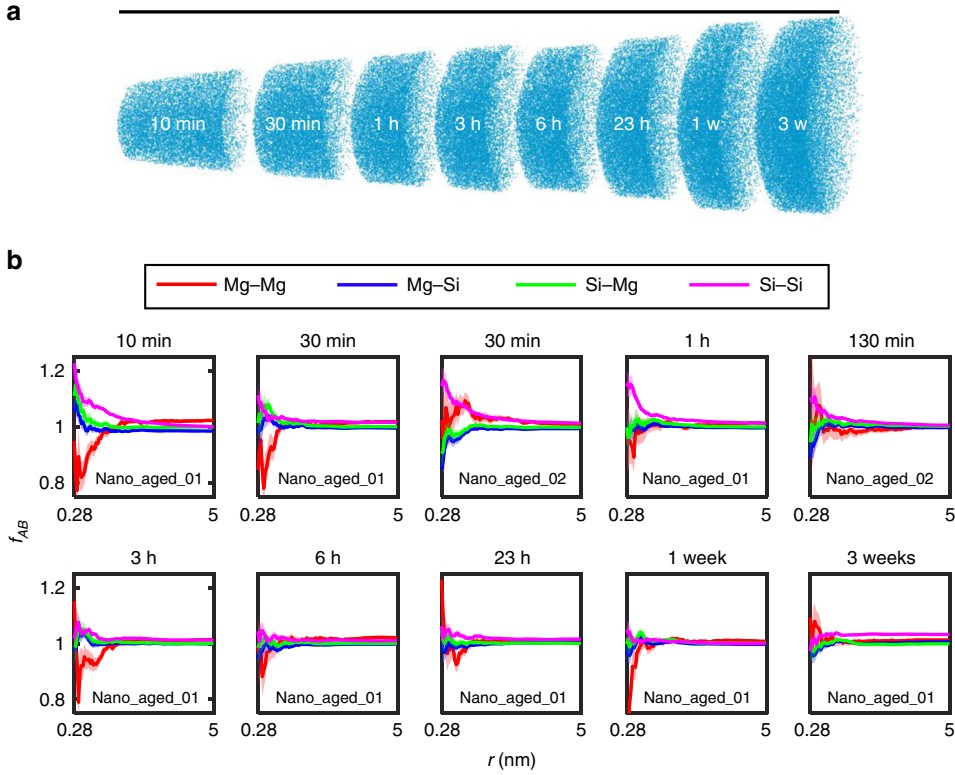

**Fig. 2** In situ natural aging of atom probe samples from a quenched AlMgSi alloy (nano aging). **a** Mg atom positions for the concatenated runs of the sample nano_aged_01 are shown to illustrate the measurement procedure. The scale bar length corresponds to 512 nm. **b** Analysis of the spatial positions of solute atoms for the samples nano_aged_01 and nano_aged_02. Shown is the ratio of the cumulative sums of the radial distribution function $f_{AB}(r)$ (Eq. (1)) for the given interactions (Mg–Mg, Mg–Si, Si–Mg, and Si–Si) and nano aging times. Values $f_{AB} > 1$ indicate clustering (for details see Supplementary Note 2 and Supplementary Table 1). No significant increase of the signals can be observed for increasing (in situ) natural aging times. Error bar boundaries are calculated according to Supplementary Eqs. (7) and (8)

can incorporate different jump frequencies for different solutes and vacancy-solute binding energies, the linear relationship between the diffusion coefficient and the vacancy concentration $c_v$ is preserved. This directly converts into the nucleation and growth rates of diffusion-controlled precipitation reactions[30].

Figure 3 shows the results of thermo-kinetic calculations (based on the method described in ref. [31]), which we conducted for a thermal route similar to the applied in situ sample processing in order to quantify the nonequilibrium vacancy fraction. The diameter of a sphere, synonymous to the maximum distance to the next vacancy sink, serves as a model for the APT specimen, which is shown in Fig. 3a. The sphere surface is modeled as ideal sink for vacancies. So are grain boundaries, which are therefore not discriminated from free surfaces[31]. The vacancy fraction calculated for pure Al is shown in Fig. 3b. Changes in the sphere diameter largely influence the evolution of the nonequilibrium vacancy fraction: the frozen-in nonequilibrium vacancy fraction upon quenching is more than an order of magnitude lower if the sphere diameter is decreased by an order, which means that even the creation of a vacancy supersaturation is difficult at small scales. The decline to the equilibrium vacancy fraction is also much earlier, and for a diameter of 100 nm, a size in the range of the APT specimen radius, it is already reached in less than a minute. The variation of the dislocation density was also tested via increasing it by a factor of $10^3$, but the nonequilibrium vacancy fraction for smaller sphere diameters did not change (see Supplementary Fig. 11b). The same applies for the use of

vacancy-binding energies of Si and Mg, which has also only a minor effect on the general conclusion (see Supplementary Fig. 11a). This suggests that the nonequilibrium vacancy-driven process of clustering must be strongly size dependent and suppressed at small dimensions.

**Bulk aging**. To further prove that clustering and NA are indeed stopped in nano-sized APT samples, we conducted a second APT experiment where the sum of "bulk aging" and "nano aging" was kept constant. The time during bulk NA (bulk aging) was varied, but the total NA time, $\Delta t_{NA}$, of the APT samples was preserved (illustrated in Fig. 4a, b). An APT sample, Fig. 4a, was prepared during 9 min after quenching of the bulk (rods of 0.7 mm thickness) and then nano aged at RT for 3 weeks. A second APT sample, Fig. 4b, was prepared after 1 week of bulk aging and then nano aged for 2 weeks. Figure 4c shows the data obtained. If the definition of $\Delta t_{NA}$ was applied, no significant difference between the two measurements would be discernible because the time at RT after quenching is 3 weeks for both runs. However, the two states differ significantly in the signal of clustering, in fact for all interactions Mg–Mg, Si–Si, Si–Mg, and Mg–Si.

Although the "nano_aged" specimen showed a decrease in Mg content of approximately 0.1% (Supplementary Table 1), likely due to Mg evaporation during the solution heat treatment, the presented bulk experiments clearly demonstrate the strong influence of the specimen size on the clustering reaction. All "bulk_aged" specimens had the same size (0.7 mm in square) at

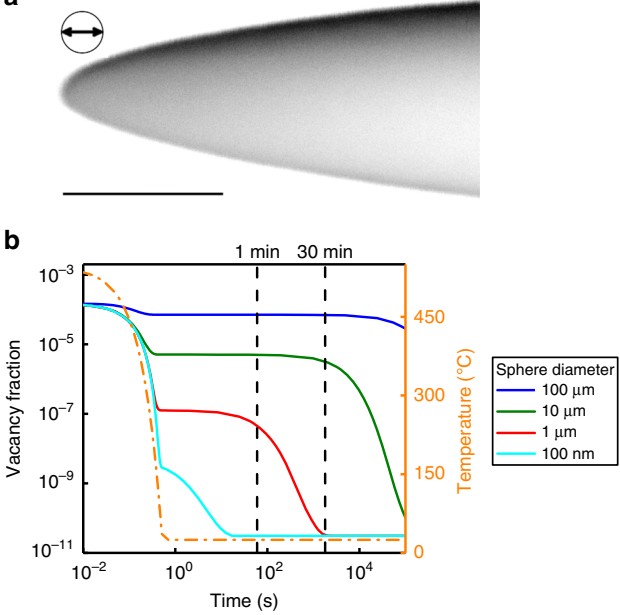

**Fig. 3** Simulation of the nonequilibrium vacancy evolution. **a** Inverted scanning electron microscopy (SEM) image of the APT sample nano_aged_01 before the measurement. The scale bar length corresponds to 500 nm and the diameter of the shown circle is 85 nm. **b** Calculated nonequilibrium vacancy fraction over time and temperature upon quenching and subsequent natural aging for pure Al (FSAK model[31]). A sphere diameter is used as a simplified model for the APT specimen. The nonequilibrium vacancy fraction formed upon quenching and its preservation at RT decays rapidly with decreasing dimensions. Additional lines for 1 min and 30 min are added as visual guidelines

the solution heat treatment and also showed almost no signal of clustering when only nano aging was applied. The small signals of 9 min bulk aging + 3 weeks nano aging, a) in Fig. 4c, may be caused by a slight clustering during the nine minutes of RT preparation at bulk dimension (Mg–Mg). However, large-scale density variations resulting from a methodical artifact may also cause such minor effects (see Supplementary Fig. 7, Si and Mg, and Supplementary Note 3). An additional nano aging of 3 weeks, i.e. from "9 min + 3 w" to "9 min + 6 w", only resulted in an insignificant change in Mg–Mg interactions (Supplementary Fig. 10), and is still significantly different from "1 w + 2 w". A replicate measurement for the same bulk aging time, but different nano aging time "1 w + 1 d" (Supplementary Fig. 10) shows results almost identical to "1 w + 2 w".

## Discussion

We have shown that clustering of solutes after quenching in metals as a diffusional process not only depends on the storage time at room temperature, as is known for NA since more than a century[5], but also significantly on sample size. The nonequilibrium diffusion rapidly stops when the sample size approaches the nanometer range. At small dimensions it is in fact impossible to retain a significant fraction of nonequilibrium vacancies upon rapid quenching, because these vacancies as the main carriers of nonequilibrium substitutional diffusion in metals are annihilated at the free surface of the nano-sized samples during quenching and storage at room temperature. This is demonstrated clearly in this study via simulations and experiments. Using nonequilibrium vacancies to accelerate solid-state reactions at a lower temperature by a solution heat treatment plus quenching strategy is thus not feasible for nano-scale objects (e.g., nanoparticles, rods or wires[32], or nanoporous alloys[33]), because the solid-state reactions for such nano-scale objects will always be dominated by their equilibrium vacancy concentrations at the applied temperature. Apart from small-scaled objects, also

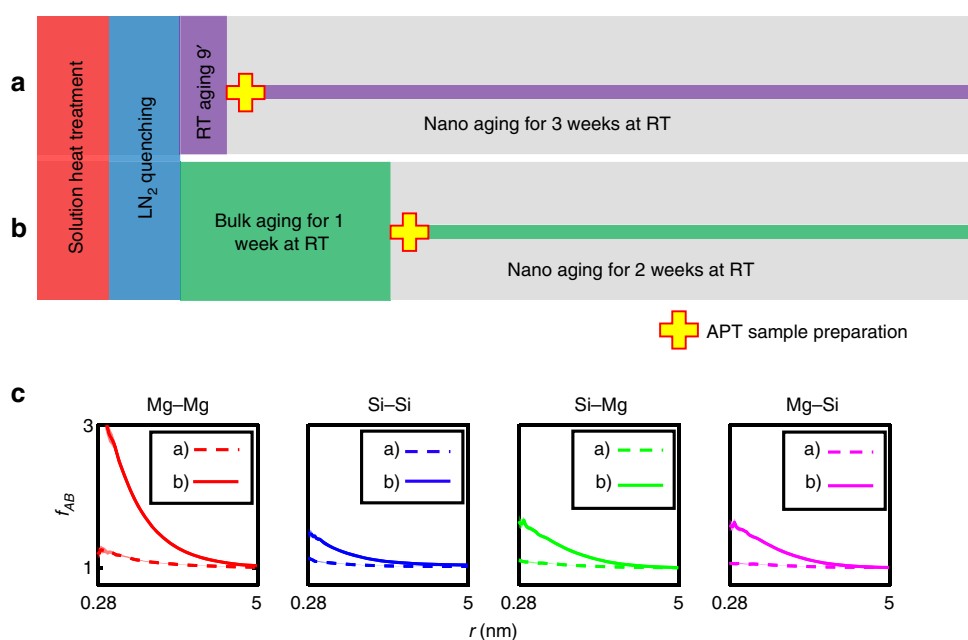

**Fig. 4** Bulk natural aging of quenched AlMgSi alloy. The schematic overview of the heat treatment and sample preparation procedure illustrates that the total time after quenching is kept constant at three weeks, while solely the time of preparation of the nano-sized atom probe specimen was varied, **a** 9 min vs. **b** 1 week. **c** Analysis of the spatial positions of the solute atoms for **a** 9 min of bulk aging and 3 weeks of nano aging and **b** 1 week of bulk aging and 2 weeks of nano aging. A pronounced difference is observed although the total time after quenching, $\Delta t_{NA}$, is similar and only the time at bulk dimensions has been varied. For the long natural aging at bulk dimensions the expected strong solute clustering upon natural aging is obvious. Error bar boundaries are calculated according to Supplementary Eqs. (7) and (8)

nanocrystalline metals[34] should exhibit these characteristics because grain boundaries also act as vacancy sinks[31]. Nanocrystalline Al alloys generated by high-pressure torsion are already present in a clustered state after processing[35], but the loss of nonequilibrium vacancies to grain boundaries requires consideration with respect to further aging. An industrially relevant process that causes grain size small enough that nonequilibrium vacancies can be quickly annihilated is friction-stir-welding[36,37]. Rapid solidification (e.g., during 3D printing of Al alloys[38,39]) also frequently causes regions of submicron-sized grains and may suppress NA and nonequilibrium vacancy-based diffusion. Thus, our findings permit several general conclusions with respect to metallic alloys.

Nonequilibrium vacancy-based substitutional diffusion-controlled processes such as solute clustering are size dependent. They are strongly suppressed at small dimensions, regardless of how the nonequilibrium vacancies were created, either by thermal quenching or other means. The size dependency presented in this study has to be considered when in situ high-resolution microscopy techniques such as TEM or APT are utilized to study nonequilibrium kinetics in bulk materials. Our findings also have specific implications on the results documented in literature concerning the observation of early stage clustering upon NA, especially in the field of Al alloys. Many results of prior applied characterization methods using small sample sizes may have been influenced by ill-defined NA time. This applies particularly to APT experiments performed over the last 20 years, and may explain differing results on short NA. For larger time spans the results are likely to hold, because such samples are usually processed contemporaneous with the APT measurements.

## Methods
Experimental details, data analysis and calculation parameters are briefly described in the following.

**Material**. A commercial EN AW 6016 Al alloy with a nominal composition of Mg 0.35%, Si 1.04%, Cu 0.04%, and Al balance (in at.%) was used for all investigations. The composition was measured using a spark optical emission spectrometer.

**Differential scanning calorimetry**. The material was cut into specimens and ground to final masses of approximately 42 mg. The specimens were heat treated at 545 °C and quenched in $LN_2$. For each NA time three specimens were measured against a high-purity Al reference of the same mass, using Al crucibles and a heating rate of 10 K min$^{-1}$. The three curves obtained were shifted to zero at the solution heat treatment regime and the mean computed. The measurements were carried out on a Netzsch DSC 204 F1.

**APT sample preparation**. Nano aging (see also Supplementary Table 1): the cut blanks ($1 \times 1 \times 20$ mm$^3$) were first-step electropolished (25% $HNO_3$ in methanol) and then a neck was micro-polished (2% $HClO_4$ in 2-butoxyethanol) near the apex towards a diameter in the order of 5–20 µm. The necked samples were then solution heat treated at 545 °C for 15 min with $N_2$ purging, quenched in $LN_2$, and transported in $LN_2$ to the arctic chamber, where the micro-polishing was completed at −40 °C (3% $HClO_4$ (72%), 16% 2-ethoxyethanol, 22% 1,2 dimethoxyethan in methanol) before further storage in $LN_2$. For NA, the $LN_2$-cooled samples were dipped into room-temperature ethanol starting the NA for the respective time (10 and 30 min for nano_aged_01 and nano_aged_02). The samples were then placed into the vacuum-cryo-specimen-transfer shuttle and the shuttle was rapidly pumped down to $10^{-6}$ mbar and the samples cooled to −120 °C using a Bal-Tec BAF060 freeze-etching chamber. The samples were subsequently vacuum-cryo transferred[27,40,41] to a FEI FIB-SEM Helios 600i device pre-cooled to −152 °C (to check the samples in SEM mode) and then to the analysis chamber of the APT. Additional NA times for the same sample were realized by stopping the run, and transferring the sample to the buffer chamber (RT) and storing it there. The respective time was added to the previous NA time.

Bulk aging (see also Supplementary Table 1): the cut blanks ($0.7 \times 0.7 \times 20$ mm$^3$) were solution heat treated at 545 °C for 15 min in an air furnace with $N_2$ purging and quenched in $LN_2$. Bulk_aged_01 (Fig. 4a) was taken out of $LN_2$ and rapidly first- and second-step electropolished within 9 min at RT. Bulk_aged_02 (Fig. 4a) and bulk_aged_03 were taken from $LN_2$ and plunged into isopropanol, stored for 1 week at room temperature, and first- and second-step electropolished. The finished samples were again stored at room temperature for 2 weeks or 1 day, respectively, until APT measurement.

**APT measurements**. The samples were run in voltage mode with a pulse fraction of 20%, a frequency of 200 kHz and a detection rate of 1% at a temperature of 30 K. The "nano_aged" samples and the sample bulk_aged_01 were run on a LEAP 4000 X HR equipped with self-constructed cryo-transfer capabilities[40,41], bulk_aged_02 and bulk_aged_03 were run on a LEAP 3000 X HR.

**APT data analysis**. For APT solute analysis the $^{24}Mg^{2+}$, $^{25}Mg^{2+}$, $^{26}Mg^{2+}$; $^{28}Si^{2+}$, $^{29}Si^{2+}$, $^{30}Si^{2+}$; and $^{24}Mg^+$, $^{25}Mg^+$, $^{26}Mg^+$ peaks were used. The reconstruction was built by calibrating the field factor $k_f$ with the observed interlayer spacing and the image compression factor with the observed angles of chosen poles[42] within the commercial program IVAS 3.6.12.

**Vacancy kinetics calulation**. A thermokinetic calculation based on the FSAK model[31], which takes excess vacancies into account, was performed using MatCalc 6. Pure Al was used as material, and the sphere diameter was varied as a model for the APT specimen diameter and vacancy sink. A temperature history of 545 °C cooled at 1000 K s$^{-1}$ to 25 °C, and further NA at 25 °C was applied. The other parameters were chosen as[2]: dislocation density $10^{11}$ m$^{-2}$, jog fraction 0.02, Frank loop nucleation constant 0.0, jog fraction on Frank loops 0.2, Frank loop interfacial energy 1.0, effective loop-line energy $1/2Gb^2$, and excess vacancy efficiency 1.0.

## Data availability
The datasets analyzed during the current study are available from the corresponding authors on reasonable request.

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

## Acknowledgements

This research was supported by the Austrian FFG Bridge project, number 853208. AMAG Rolling GmbH is thanked for financial support and discussions. This project also received funding from the European Research Council (ERC) under the European Union's Horizon 2020 research and innovation program (grant No. 757961).

The authors thank Mr. Camenzind, Mr. Rechsteiner and Mr. Eisenhut from EMPA, and Mr. Jaggi and Mr. Schneebeli from SLF Davos for organization and help in using their arctic chambers. We also kindly thank Mr. Kollender from JKU for his recommendation of an electrolyte suitable for electro-polishing at low temperatures; Mr. Bartelme of Montanuniversitaet Leoben for his help with the sample production in the arctic chamber; Mr. Tunes, Montanuniversitaet Leoben, for his help in designing the figures; and Ms. Mendez Martin, Montanuniversitaet Leoben, for organizing APT measurements.

## Author contributions

P.D., S.P., and P.J.U. conceived the study. P.D. and P.J.U. produced the samples, and P.D. and S.S.A.G. performed the measurements. P.D. and S.P. did the calculations, and P.D. analyzed the data. S.P., P.J.U., and J.F.L. coordinated and supervised the work. All authors extensively discussed the data. P.D. wrote the paper with the support and correction of all other authors.

## Competing interests

The authors declare no competing interests.
