## [Peer Review File · Nature Communications]

Reviewers' comments:

Reviewer #1 (Remarks to the Author):

This is a nice paper showing the importance of sample size upon natural ageing at room temperature. The article is clearly written and experiments properly supports conclusions. Apart from a few typing errors in references (e.g. almgSi instead of AlMgSi), I would suggest to more clearly explicit the role of vacancy supersaturation on diffusion of solute elements and on nucleation rate of precipitates.

I also suggest to extend the discussion to the influence of - not only the sample size and surface elimination of supersaturated vacancies - but also that of grain size or even dislocations when samples are submitted to severe deformation. Dislocations and grain boundaries (GBs) are also important defect sinks in the same way as sample surfaces. It seems quite clear that nano-sized grains will give rise to more rapid elimination of vacancies to GBs and this will rapidly reduce the diffusion rate of solute elements and consequently the nucleation rate of precipitation.

Reviewer #2 (Remarks to the Author):

The manuscript reports on the suppression of clustering during so-called natural ageing in an Al-based alloy because of the annihilation of vacancies in an object that has sub-micron dimensions. This appears clearly in the comparison to a bulk sample that has been subject to natural ageing for a similar time and that exhibits strong signal in the pair-correlation functions calculated from atom probe tomography data.

If the case is rather convincing, the experiments are surely challenging yet the paper doesn't feel strong enough, the figures not very clear, the discussion weak. I think this stems from the fact that the paper is too short and likely just a transfer form one of the less general Nature-branded journals.

Some critical points hence aren't discussed to the level of details that they should.

For instance, why is the 20% increase at short distance in the Si-Si PDF not seemed significant? the difference in behaviour between the nano-NA and the bulk-NA samples seem to be mostly between Mg and Si, why?

The shape of the samples for the solution heat-treatment was quite different in both cases, would this not have an influence?

Then we're left with the question of how general is this process actually is... NA is mostly only critical in Al-based alloys. Maybe in nano-objects too, but this is once again not really discussed.

I think the results are nice and impressive, I'm just worried that the way they are presented does not do them justice and the readers will be left with many questions...

more specific details can be found in the attached file.

Reviewer #3 (Remarks to the Author):

This manuscript "Size Dependent Diffusion: Material Dimensions Determine Solid State Reactions" by Dumitraschkewitz et al. describes a set of very interesting experiments on room temperature clustering in an aluminium alloy.

By using their original cryo-transfer and cryo-sample-preparation setup, the authors have

managed to use a destructive high-resolution technique (atom probe tomography, APT) for near-in-situ experiments, by performing interrupted natural ageing of an APT sample in the storage chamber, with interrupted APT analyses in between.

These results show that no clustering occurs, in contrast to what is observed in their ex situ samples where natural ageing occurred on bulk samples.

This is a quite remarkable result, which the authors quite convincingly explain by an accelerated annihilation of the quenched-in vacancies at the surface of the tip, compared to bulk samples (where vacancy sinks are less numerous).

I must say I am a little less convinced by the generality of this result which is put forward by the authors, as evidenced by the very general title and some of the conclusions. For instance, in the conclusions " Our findings permit several general statements regarding metal alloys. All non-equilibrium substitutional diffusion-controlled processes are size dependent. They are strongly suppressed at small dimensions, regardless of how non-equilibrium vacancies are created, by thermal quenching or other means.": I don't believe this particular experiment enables to draw this very general conclusion.

This is a particular situation where the annihilation of vacancies is key to control the rate of the reaction. But this is not at all a general case. While vacancies are indeed what controls the diffusion in a substitutional alloy, the concentration of vacancies usually reaches its equilibrium concentration quite fast (vacancies are fast diffusers). This is what made the discovery of Alfred Wilm surprising and is particular to clustering in Al alloys. Of course, I agree that diffusion may be size dependent, and this is an example where it is, but this example does not apply to all solid state reactions, as is implied by the title.

All in all, the need for broad applicability and general conclusions (maybe due to the broad readership aimed by Nature Comm.) has probably prevented the authors to discuss more the extremely interesting and helpful consequences this work has on the clustering in aluminium alloys specifically.

Some other comments:

- There are other differences in the ageing history of the the samples, such as the fact that the "nano-tip" samples were solution treated and quenched as fine (few μm) pre-polished samples whereas the bulk one were significantly bigger. This could have resulted in a loss of solutes (and hence, in less clustering). This should have been discussed in the text. I spotted a table in supplementary materials suggesting that, indeed, the nano-tip have less solutes than the bulk aged. Also, the SHT time should be mentioned.

- The description of the function used to analyse APT data in terms of clustering is slightly unclear. I am not sure I fully grasped what the authors mean by "ratio of the cumulative sums of the radial distribution functions" (p.9) and I found the supplementary materials very helpful in this aspect. The notion of "cumulative sum" is quite nebulous here. There is a quote to ref. [25] where the authors actually do not seem to use radial distribution functions, but a ratio based on the 10th nearest neighbour distribution.

- Fig. 2b is not very legible, but it can not be said that no clustering occurs and that the distribution of solute is completely random. The Si-Si in particular significantly increases (albeit clearly less than on the bulk aged samples). What is even more striking is the difference of the Mg-Mg behaviour between nano and bulk. This would have deserved more discussion.

Frédéric De Geuser

Reviewer #1

Reviewer #1: This is a nice paper showing the importance of sample size upon natural ageing at room temperature. The article is clearly written and experiments properly supports conclusions.

Authors: We thank the reviewer for the positive feedback on the manuscript.

Reviewer #1: Apart from a few typing errors in references (e.g. almgSi instead of AlMgSi), I would suggest to more clearly explicit the role of vacancy supersaturation on diffusion of solute elements and on nucleation rate of precipitates.

I also suggest to extend the discussion to the influence of - not only the sample size and surface elimination of supersaturated vacancies - but also that of grain size or even dislocations when samples are submitted to severe deformation. Dislocations and grain boundaries (GBs) are also important defect sinks in the same way as sample surfaces. It seems quite clear that nano-sized grains will give rise to more rapid elimination of vacancies to GBs and this will rapidly reduce the diffusion rate of solute elements and consequently the nucleation rate of precipitation.

Authors: We have corrected the typing errors for chemical elements in the references, which unfortunately occurred due to the automatic use of lower case characters.

We have added text in the chapter Results, Vacancy annihilation, describing the general influence of vacancies on the diffusion coefficients of elements and its consequent impact on precipitation:

“In general, diffusion of a pure element by a random walk of vacancies can be described by

$$D = \frac{1}{6} a^2 n c_v \omega , (2)$$

where D is the self-diffusion coefficient, a denotes the lattice parameter, n is the coordination number, c_v the vacancy concentration and ω is the jump frequency of the vacancy [29]. c_v directly influences the self-diffusion coefficient. While the description of solute diffusion in a matrix is more complex and can incorporate different jump frequencies for different solutes and vacancy-solute binding energies, the linear relationship between the diffusion coefficient and the vacancy concentration c_v is preserved. This directly converts into the nucleation and growth rates of diffusion-controlled precipitation reactions [30].” (page 5 and 6)

We further describe now that the used vacancy annihilation model does not discern between a grain boundary or a free surface:

“The sphere surface is modelled as ideal sink for vacancies. So are grain boundaries, which are therefore not discriminated from free surfaces [31].” (page 6)

We also tested now this vacancy annihilation model for an increase in dislocation density and describe the results; we added an additional diagram to the Supplementary:

“The variation of the dislocation density was also tested via increasing it by a factor of 10^3 , but the non-equilibrium vacancy fraction for smaller sphere diameter did not change (see Supplementary Fig. 11b). The same applies for the use of vacancy-binding energies for Si and Mg, which has also only a minor effect on the general conclusion (see Supplementary Fig. 11a).” (page 6)

Additionally, we added a discussion on potential effects of our discovery on nanocrystalline material in the discussion section:

“Apart from small-scaled objects, also nanocrystalline metals [34] should exhibit these characteristics as grain boundaries also act as vacancy sinks [31]. Nanocrystalline Al alloys generated by high-pressure torsion are already present in a clustered state after processing [35], but the loss of non-equilibrium vacancies to grain boundaries requires consideration with respect to further aging. An industrially relevant process that causes grain size small enough that non-equilibrium vacancies can be quickly annihilated is friction-stir-welding [36, 37]. Rapid solidification (e.g. during 3D printing of Al alloys [38, 39]) also frequently causes regions of sub-micron sized grains and may suppress natural aging and non-equilibrium vacancy-based diffusion.” (page 8 and 9)

Reviewer #2

Reviewer #2: The manuscript reports on the suppression of clustering during so-called natural ageing in an Al-based alloy because of the annihilation of vacancies in an object that has sub-micron dimensions. This appears clearly in the comparison to a bulk sample that has been subject to natural ageing for a similar time and that exhibits strong signal in the pair-correlation functions calculated from atom probe tomography data.

Authors: We thank the reviewer for the positive feedback on the content of the manuscript.

Reviewer #2: If the case is rather convincing, the experiments are surely challenging yet the paper doesn't feel strong enough, the figures not very clear, the discussion weak. I think this stems from the fact that the paper is too short and likely just a transfer form one of the less general Nature-branded journals. Some critical points hence aren't discussed to the level of details that they should.

Authors: The reviewer is right that the paper was initially the paper was designed as a short paper and was rewritten before submission. We now have added additional discussion, but want to keep the paper terse. Therefore the technical details are described in-depth in the Supplementary, but well linked with the paper. The main conclusions of the technical details are now better embedded in the main text. We added additional discussion and respond to the points raised in the following.

Reviewer #2: For instance, why is the 20% increase at short distance in the Si-Si RDF not seemed significant? The difference in behaviour between the nano-NA and the bulk-NA samples seem to be mostly between Mg and Si, why?

Authors: We attribute 20% increase to a field-evaporation artefact and added the following comment to the main text:

“A trend that may be visible is a decrease in Si-Si interactions with NA, which is contrary to the expected trend for clustering. This can be attributed to a field-evaporation artefact of Si rather than to an increased natural aging time (see Supplementary Note 3). (page 5)”

Due to extensive text for reasoning this argument and due to the fact that it is not the main objective of this paper, this technical detail of the characterization method is given to the Supplementary. Further additional text in the Supplementary Note 3 was added to avoid misinterpretation.

Reviewer #2: The shape of the samples for the solution heat-treatment was quite different in both cases, would this not have an influence?

Authors: We obtained a decrease in Mg content for “nano_aged” specimen, likely due to evaporation during solution heat treatment affecting samples with small dimensions. We now explicitly refer to this artefact in the section Results, Bulk aging, and reason that alone with the “bulk_aged” samples the same effect is obtained:

“Although the ”nano aged” specimen showed a decrease in Mg content of approx. 0.1% (Supplementary Table 1), likely due to Mg evaporation during the solution heat treatment, the presented bulk experiments clearly demonstrate the strong influence of the specimen size on the clustering reaction. All ”bulk aged” specimen had the same size (0.7 mm in square) at the solution heat treatment and also showed almost no signal for clustering when nano aging was applied. The small signals of (9’ + 3 weeks) in Fig. 4b may be caused by a slight clustering during the nine minutes of RT preparation at bulk dimension. However, large-scale density variations could also cause such an effect (see Supplementary Fig. 7 and Supplementary Note 3). An additional nano aging of 3 weeks, i.e. from (9’ + 3 weeks) to (9’ + 6 weeks), only resulted in an insignificant change in Mg-Mg interactions (Supplementary Fig. 10), and is still significantly different from (1 week + 2 weeks).” (page 7)

The very challenging experiments with the “nano_aged” samples is a forerunner of this conclusion.

Reviewer #2: Then we're left with the question of how general is this process actually is... NA is mostly only critical in Al-based alloys. Maybe in nano-objects too, but this is once again not really discussed.

Authors: For substitutional diffusional processes which depend on non-equilibrium vacancies, there is no argument against the generality of the found effect which the authors are aware of.

We added additional discussion for possible implications for nano-scale objects:

“Using non-equilibrium vacancies to accelerate solid-state reactions at a lower temperature by a solution heat treatment – quenching strategy is not feasible for nano-scale objects (e.g. nanoparticles, rods or wires [32] or nanoporous alloys [33]), because the solid-state reactions for such nano-scale objects will always be dominated by their equilibrium vacancy concentrations at the applied temperature.” (page 8)

And further as mentioned above (see reviewer 1) for nanocrystalline material:

“Apart from small-scaled objects, also nanocrystalline metals [34] should exhibit these characteristics as grain boundaries also act as vacancy sinks [31]. Nanocrystalline Al alloys generated by high-pressure torsion are already present in a clustered state after processing [35], but the loss of non-equilibrium vacancies to grain boundaries requires consideration with respect to further aging. An industrially relevant process that causes grain size small enough that non-equilibrium vacancies can be quickly annihilated is friction-stir-welding [36, 37]. Rapid solidification (e.g. during 3D printing of Al alloys [38, 39]) also frequently causes regions of sub-micron sized grains and may suppress natural aging and non-equilibrium vacancy-based diffusion.” (page 8 and 9)

Reviewer #2: I think the results are nice and impressive, I'm just worried that the way they are presented does not do them justice and the readers will be left with many questions... more specific details can be found in the attached file.

Authors: We thank the author for the positive valuation of the results. We hope to have improved the presentation of the results satisfactorily with the carefully revised manuscript. We correspond to the specific details in the sent file in the following:

Reviewer #2 [A1]: Sounds odd...Relates to?

Authors: Text changed: “implies” was exchanged by “*relates to*”

Reviewer #2 [A2]: Only for substitutional elements. Please be specific

Authors: Text changed: “*In metals, kinetics is primarily connected to diffusion, which for substitutional elements is enabled via mobile atomic-lattice vacancies.*”

Reviewer #2 [A3]: Examples? Be specific

Authors: Text changed: “*Rapid quenching of various important alloys, such as Al- or Mg-alloys, ...*”

Reviewer #2 [A4]: I would argue that APT is not an imaging method. It's a mass spectrometry technique with high spatial resolution

Authors: As we always analyze 3D data which represent a map of the elemental distribution in a specimen, we consider it justified to call atomic tomography an imaging method.

Reviewer #2 [A5/A6]: I would avoid brackets in an abstract, vague

Authors: Text changed: "... which, besides its technological and academic importance, ...".

Reviewer #2 [A7]: by

Authors: Text changed: "via" to "by"

Reviewer #2 [A8]: ? reads strange

Authors: Text changed: "Already in 1911 Wilm [5] discovered hardening of Al alloys during room-temperature storage when trying to increase the hardness of an Al alloy in a procedure similar to steel quenching."

Reviewer #2 [A9]: Is there not a reference to support this strong statement?

Authors: Text changed: "*The kinetics of NA strongly depends on non-equilibrium vacancies [2], ...*"

Reviewer #2 [A10/A11]: And microanalysis / I would argue that even APT only has near-atomic and tbh more likely sub-nanometer resolution

Authors: Text changed: "... characterization methods, i. e. microscopy and microanalysis techniques with atomic (transmission electron microscopy, TEM) and near atomic (atom probe tomography, APT) resolution [10, 11, 12, 13].

Reviewer #2 [A12]: Please be consistent with naming, either Al-alloys or aluminium alloys. Also please use British spelling

Authors: We changed aluminium/aluminum to Al in the whole text.

Reviewer #2 [A13]: Which ones?

Authors: Text changed: "*However, it should be also noted that NA can be also beneficial for gaining strength in other low-alloyed AlMgSi alloys [17], Mg alloys [18] or AlCuMg alloys [6].*"

Reviewer #2 [A14]: This sounds like APT jargon that I would think should be avoided

Authors: We avoid now the term "tip" in the whole text, Figures and Tables.

Reviewer #2 [A15]: No no no no, Needle-shaped specimens

Authors: Text changed: “*The first uses ”nano aging” where the NA is performed in-situ in the atom probe on finished needle-shaped specimens.*”

Reviewer #2 [A16]: colloquial

Authors: The phrase “To our surprise and excitement” was neglected.

Reviewer #2 [A17/A18]: well there is a 20% increase of likelihood to find Si near Si which might well be significant / this is far from obvious from looking at the data it is much less than in the second case, admittedly!

Authors: We attribute 20% increase (nano_aged_01, 10’) to a field-evaporation artefact (as already discussed above).

We reformulated the two sentences: “*Spatial analyses of the solute species Mg and Si, shown in Fig. 2b, reveal no significant increase for the Mg-Mg, Si-Mg, Mg-Si or Si-Si interactions due to in-situ aging (see also Supplementary Fig. 2). The solute distribution only differs slightly from a random comparator and does not increase over the applied NA time, Δt_{NA} , up to three weeks. A trend that may be visible is a decrease in Si-Si interactions with NA, which is contrary to the expected trend for clustering. This can be attributed to a field-evaporation artefact of Si rather than to an increased natural aging time (see Supplementary Note 3).*”

Reviewer #2 [A19]: A sphere is a poor model for a needle. Please don’t use tip. Also why not refer to this as a sphere afterwards – this would make much more sense

Authors: While it is true that a sphere is a poor model for a needle, it is the only model we have available and we consider it sufficient for our estimation.

Reviewer #2 [A20/21]: I think beyond a few microscopy techniques, this could have implication on nano-scale objects, ie nanoparticles, nanowires etc. This should be discussed. But how relevant is this discussion to industrial alloys, likely not so relevant... all are bulky compared to a sub-micron sized sample

Authors: We added additional discussion for possible implications for nano-scale objects:

“*Using non-equilibrium vacancies to accelerate solid-state reactions at a lower temperature by a solution heat treatment – quenching strategy is not feasible for nano-scale objects (e.g. nanoparticles, rods or wires [32] or nanoporous alloys [33]), because the solid-state reactions*”

for such nano-scale objects will always be dominated by their equilibrium vacancy concentrations at the applied temperature.” (page 8)

And further for nanocrystalline material:

“Apart from small-scaled objects, also nanocrystalline metals [34] should exhibit these characteristics as grain boundaries also act as vacancy sinks [31]. Nanocrystalline Al alloys generated by high-pressure torsion are already present in a clustered state after processing [35], but the loss of non-equilibrium vacancies to grain boundaries requires consideration with respect to further aging. An industrially relevant process that causes grain size small enough that non-equilibrium vacancies can be quickly annihilated is friction-stir-welding [36, 37]. Rapid solidification (e.g. during 3D printing of Al alloys [38, 39]) also frequently causes regions of sub-micron sized grains and may suppress natural aging and non-equilibrium vacancy-based diffusion.” (page 8 and 9)

Also we think it is important for the analysis of bulk materials with analytical methods operating on nano-scale samples as already mentioned in the discussion section.

Reviewer #2 [A22]: Why? Why are they placed in the FIB? What temperature is the FIB stage?

Authors: We used the cryo-FIB in SEM mode to check the specimen.

Text changed: *The samples were subsequently vacuum-cryo transferred [40, 41, 27] to a FEI FIB-SEM Helios 600i device pre-cooled to -152° C (to check the samples in SEM mode) ...”*

Reviewer #2 [A23]: Why not with similar neck? The change in volume at the neck could also play a role with respect to the vacancy concentration, but also to the possible loss of e.g. Mg during the heat treatment etc.

Authors: “Bulk_aged” specimen are intended to have a bulk size during solution heat treatment: generally to be able to natural age the specimen at bulk dimension and also to avoid possible Mg loss during solution heat treatment. Further to avoid loss of vacancies during quenching as already discussed (Discussion) in the manuscript: *“At small dimensions it is in fact impossible to retain a significant fraction of non-equilibrium vacancies upon rapid quenching, because these vacancies as the main carriers of non-equilibrium substitutional diffusion in metals are annihilated at the free surface of the nano-sized samples during quenching and storage at room temperature.” (page 8)*

Reviewer #3

Reviewer #3: This manuscript "Size Dependent Diffusion: Material Dimensions Determine Solid State Reactions" by Dumitraschkewitz et al. describes a set of very interesting experiments on room temperature clustering in an aluminium alloy.

By using their original cryo-transfer and cryo-sample-preparation setup, the authors have managed to use a destructive high-resolution technique (atom probe tomography, APT) for near-in-situ experiments, by performing interrupted natural ageing of an APT sample in the storage chamber, with interrupted APT analyses in between.

These results show that no clustering occurs, in contrast to what is observed in their ex situ samples where natural ageing occurred on bulk samples.

This is a quite remarkable result, which the authors quite convincingly explain by an accelerated annihilation of the quenched-in vacancies at the surface of the tip, compared to bulk samples (where vacancy sinks are less numerous).

Authors: The reviewer is kindly thanked for the positive feedback on our results and seeing the importance of our finding.

Reviewer #3: I must say I am a little less convinced by the generality of this result which is put forward by the authors, as evidenced by the very general title and some of the conclusions. For instance, in the conclusions " Our findings permit several general statements regarding metal alloys. All non-equilibrium substitutional diffusion-controlled processes are size dependent. They are strongly suppressed at small dimensions, regardless of how non-equilibrium vacancies are created, by thermal quenching or other means.": I don't believe this particular experiment enables to draw this very general conclusion.

Authors: We amended the title, to avoid a too general statement, to "*Size-Dependent Diffusion: A Solid-State Reaction Determined by Material Dimension*".

Further we changed the mentioned statement to:

"Thus, our findings permit several general conclusions regarding metal alloys.

All non-equilibrium vacancy-based substitutional diffusion-controlled processes are size dependent. They are strongly suppressed at small dimensions, regardless of how the non-equilibrium vacancies are created, either by thermal quenching or other means." (page 8 and 9)

This is just a general physical conclusion based on the obtained effect. The authors are not aware of an argument, why this behavior should change for other metallic systems. However, as also

requested by reviewer #1 we added a comment on the importance of the vacancy concentration for diffusion:

“In general, diffusion of a pure element by a random walk of vacancies can be described by

$$D = \frac{1}{6} a^2 n c_v \omega , (2)$$

where D is the self-diffusion coefficient, a denotes the lattice parameter, n is the coordination number, c_v the vacancy concentration and ω is the jump frequency of the vacancy [29]. c_v directly influences the self-diffusion coefficient. While the description of solute diffusion in a matrix is more complex and can incorporate different jump frequencies for different solutes and vacancy-solute binding energies, the linear relationship between the diffusion coefficient and the vacancy concentration c_v is preserved. This directly converts into the nucleation and growth rates of diffusion-controlled precipitation reactions [30]. (page 5 and 6)

Moreover, again in line with the suggestions from reviewer #1 and #2 we discuss now other specific cases in the where the effect is of importance:

“Using non-equilibrium vacancies to accelerate solid-state reactions at a lower temperature by a solution heat treatment – quenching strategy is not feasible for nano-scale objects (e.g. nanoparticles, rods or wires [32] or nanoporous alloys [33]), because the solid-state reactions for such nano-scale objects will always be dominated by their equilibrium vacancy concentrations at the applied temperature.” (page 8)

and

“Apart from small-scaled objects, also nanocrystalline metals [34] should exhibit these characteristics as grain boundaries also act as vacancy sinks [31]. Nanocrystalline Al alloys generated by high-pressure torsion are already present in a clustered state after processing [35], but the loss of non-equilibrium vacancies to grain boundaries requires consideration with respect to further aging. An industrially relevant process that causes grain size small enough that non-equilibrium vacancies can be quickly annihilated is friction-stir-welding [36, 37]. Rapid solidification (e.g. during 3D printing of Al alloys [38, 39]) also frequently causes regions of sub-micron sized grains and may suppress natural aging and non-equilibrium vacancy-based diffusion.” (page 8 and 9)

Reviewer #3: This is a particular situation where the annihilation of vacancies is key to control the rate of the reaction. But this is not at all a general case. While vacancies are indeed what controls the diffusion in a substitutional alloy, the concentration of vacancies usually reaches its

equilibrium concentration quite fast (vacancies are fast diffusers). This is what made the discovery of Alfred Wilm surprising and is particular to clustering in Al alloys. Of course, I agree that diffusion may be size dependent, and this is an example where it is, but this example does not apply to all solid state reactions, as is implied by the title.

Authors: Indeed, this effect will not occur for reactions which operate at equilibrium vacancy concentration. To avoid misinterpretation we further concretized our formulation:

*“All non-equilibrium **vacancy-based** substitutional diffusion-controlled processes are size dependent. They are strongly suppressed at small dimensions, regardless of how **the** non-equilibrium vacancies are created, **either** by thermal quenching or other means.”*

Also we amended the title as described above.

Reviewer #3: All in all, the need for broad applicability and general conclusions (maybe due to the broad readership aimed by Nature Comm.) has probably prevented the authors to discuss more the extremely interesting and helpful consequences this work has on the clustering in aluminium alloys specifically.

Authors: Many thanks for seeing the importance of the effect for already aluminium alloys. However, we tried to aim the paper on the effect of the size-dependence and not specifically on the characterization of the clustering, although this is also of great interest for the aluminum community. To account for this we are more explicit now on that in the abstract:

*“Rapid quenching of various important alloys, **such as Al- and Mg-alloys, ...**” and “We illustrate the size effect on clustering in an **AlMgSi...**”*

We already stated that for short natural aging times, the aging time has possibly been ill-defined for earlier studies and highlight its consequences for natural aging of aluminium alloys in the conclusions. Further we now strengthened the discussion of the effect of choosing the region of interest on the “non-randomness” and field-evaporation artefacts, which is of interest for the APT community working on clustering, in the Supplementary Information and better linked this to the manuscript (see response to reviewer #2).

Some other comments:

Reviewer #3: There are other differences in the ageing history of the samples, such as the fact that the “nano-tip” samples were solution treated and quenched as fine (few μm) pre-polished samples whereas the bulk one were significantly bigger. This could have resulted in a loss of solutes (and hence, in less clustering). This should have been discussed in the text. I spotted a table in

supplementary materials suggesting that, indeed, the nano-tip have less solutes than the bulk aged. Also, the SHT time should be mentioned.

Authors: We obtained a decrease in Mg content for “nano_aged” specimen, likely due to evaporation during solution heat treatment affecting samples with small dimensions. We now explicitly refer to this artefact in the section results, bulk aging, and reason that alone with the “bulk_aged” samples the same effect is obtained:

“Although the ”nano aged” specimen showed a decrease in Mg content of approx. 0.1% (Supplementary Table 1), likely due to Mg evaporation during the solution heat treatment, the presented bulk experiments clearly demonstrate the strong influence of the specimen size on the clustering reaction. All ”bulk aged” specimen had the same size (0.7 mm in square) at the solution heat treatment and also showed almost no signal for clustering when nano aging was applied. The small signals of (9’ + 3 weeks) in Fig. 4b may be caused by a slight clustering during the nine minutes of RT preparation at bulk dimension. However, large-scale density variations could also cause such an effect (see Supplementary Fig. 7 and Supplementary Note 3). An additional nano aging of 3 weeks, i.e. from (9’ + 3 weeks) to (9’ + 6 weeks), only resulted in an insignificant change in Mg-Mg interactions (Supplementary Fig. 10), and is still significantly different from (1 week + 2 weeks).” (page 7)

The very challenging experiments with the “nano_aged” samples is a forerunner of this conclusion. We added the solution heat treatment time of “15 min” for nano_aged and bulk_aged samples in the methods section, we apologize for this missing important information.

Reviewer #3: The description of the function used to analyse APT data in terms of clustering is slightly unclear. I am not sure I fully grasped what the authors mean by "ratio of the cumulative sums of the radial distribution functions" (p.9) and I found the supplementary materials very helpful in this aspect. The notion of "cumulative sum" is quite nebulous here. There is a quote to ref. [25] where the authors actually do not seem to use radial distribution functions, but a ratio based on the 10th nearest neighbour distribution.

Authors: We enhanced equation 1 to better understand what is meant. Further the following comment is added to the section Results, Nano aging, before the first spatial analyses are discussed:

“We use the ratio of the cumulative sums of the radial distribution function (RDF) for interactions of species A and B (AB) for spatial analysis (equation 1 [26]), where values > 1 indicate clustering.

$$f(r) = \frac{\sum_0^r RDF_{AB}(R)}{\sum_0^r RDF_{AB,random}(R)} (1)$$

This formalism has the advantage of being parameter-free [28], in comparison to a clusterfinding algorithm which strongly depends on the input parameters, and still comprises the information from the whole spatial distribution of chosen solutes within a given radius r . A more detailed description of the analysis method can be found in the Supplementary Note 2.

(page 4 and 5)

A more detailed description is kept in the Supplementary Information in order to simplify reading. In the mentioned reference (previously [25]) the 10th NN distributions in the upper parts of the figures and in the lower parts, the same formalism as we use it here (equation 1) is used and is labeled as “ratio” in the figures.

Reviewer #3: Fig. 2b is not very legible, but it can not be said that no clustering occurs and that the distribution of solute is completely random. The Si-Si in particular significantly increases (albeit clearly less than on the bulk aged samples). What is even more striking is the difference of the Mg-Mg behavior between nano and bulk. This would have deserved more discussion.

Authors: Fig. 2b is intended to be directly comparable to Fig. 4b and therefore has the same bounds. However the data is magnified in Supplementary Fig. 2.

The small signal of the Si-Si is decreasing for the different measured times. We attribute this to a field-evaporation artefact and added a comment on this:

“A trend that may be visible is a decrease in Si-Si interactions with NA, which is contrary to the expected trend for clustering. This can be attributed to a field-evaporation artefact of Si rather than to an increased natural aging time (see Supplementary Note 3).” (page 5)

Moreover, this is analyzed in an improved Supplementary Note 3: Surface migration/surface relaxation and regions of interest.

The difference of the Mg-Mg behaviour between nano and bulk is also addressed in the section results, bulk aging as already described above.

Reviewers' comments:

Reviewer #1 (Remarks to the Author):

The article can now be considered for publication

Reviewer #2 (Remarks to the Author):

The authors made a bit of an effort to improve the readability of their manuscript but I still tend to disagree with some of their points and I think they ignored some of my main criticisms. Overall, I don't think the authors have vastly improved on their initial version.

Again, the figures look poor, really hard to follow, those in the suppl. are worse... Suppl. Fig 10 or 11 have legend overlapping with the plotted data, Suppl. Fig 2 has tons of data from which we can barely distinguish anything... it is even difficult to see any trends in there. I don't know why so little attention was paid to this.

I'm not convinced by some of their answers

"A trend that may be visible is a decrease in Si-Si interactions with NA, which is contrary to the expected trend for clustering. This can be attributed to a field-evaporation artefact of Si rather than to an increased natural aging time (see Supplementary Note 3). (page 5)"

but their Fig 4 shows the exact opposite, with an increase in Si-Si pair correlations at short distances... so what to believe? this is very confusing and was not addressed despite my comments.

The discussion on the artefact in the APT and associated surface diffusion is really unclear - there are arguments that the increase in Si composition at poles and zone lines is related to chromatic aberrations (see Marquis and Vurpillot, M&M 2008) affecting the Al preferentially, which leads to a local loss of Al and not an increased amount of Si. The paper by Gault et al. mostly demonstrates this for interstitials that can easily jump from one atomic position to another, which is unlikely to be the case between Si and Al, and the mechanism proposed by Oberdorfer et al. relies on the use of potentials that are likely ill-defined for surfaces and on time-scales that may not be realistic. If at play, all the solutes would always be on zone lines! it's not the case. These mechanisms may not be at play at all, and for now, I don't see how the author's argument is convincing or advancing understanding on these fronts. I know this is likely beyond the scope of this paper, but I feel that the authors are using arguments that sound scientific to make a specific point fit their story.

Reviewer #3 (Remarks to the Author):

I acknowledge that the authors have responded to each comment. In particular, they have attempted to make discuss the generality of the effect more deeply and have amended the title accordingly, which is appreciated.

This being said, I still believe that these nice experiments are less broadly relevant than claimed, the influence of non-equilibrium quenched-in vacancies on further kinetics concerning mostly aluminium alloys engineering. What the authors have highlighted here is that:
- the annihilation rate of non-equilibrium quenched-in vacancies depends on the density of vacancy

"sinks" and annihilation sites (which is, of course, not surprising, but these experiments are a nice way to actually measure it).

- these type of in situ (or pseudo in situ) experiments at this scale have limited representativity with respect to what happens in the bulk material. This is similar to what happens with in situ TEM ageing experiments where surface effects are important

I still believe that presenting these results along those lines would have a much higher impact on the community, rather than trying to write them very generally.

Reviewer #1

Reviewer #1: The article can now be considered for publication.

Authors: We thank the reviewer for the positive evaluation of the revision of the manuscript.

Reviewer #2

Reviewer #2: The authors made a bit of an effort to improve the readability of their manuscript but I still tend to disagree with some of their points and I think they ignored some of my main criticisms. Overall, I don't think the authors have vastly improved on their initial version.

Again, the figures look poor, really hard to follow, those in the suppl. are worse... Suppl. Fig 10 or 11 have legend overlapping with the plotted data, Suppl. Fig 2 has tons of data from which we can barely distinguish anything... it is even difficult to see any trends in there. I don't know why so little attention was paid to this.

Authors: We are very sorry that Reviewer #2 was not convinced by the quality of the figures in the manuscript. For Suppl. Fig. 2 it was maybe too enthusiastic to put all 80 curves into such a small figure. This was made to increase the transparency and traceability of the data analysis but unfortunately resulted in the opposite.

We now intensively worked on an overall improvement of readability and quality of the figures. (See Figure 1, 2, 3 and 4 of the manuscript and Suppl. Fig. 1, 2, 6, 10 and 11).

Reviewer #2: I'm not convinced by some of their answers

"A trend that may be visible is a decrease in Si-Si interactions with NA, which is contrary to the expected trend for clustering. This can be attributed to a field-evaporation artefact of Si rather than to an increased natural aging time (see Supplementary Note 3). (page 5)"

Authors: The addressed small trend was really hard to see in the old figure and is much clearer in the revised figure. We also improved the mentioned sentence to "A trend that may be visible is a decrease in Si-Si interactions with *in-situ* natural aging after the "10 min" NA measurement (Fig. 2), which is contrary to the expected trend for clustering. This may be attributed to the method specific field-evaporation artefact of Si rather than to an increased natural aging time (see Supplementary Note 3). (page 5)" ...

Reviewer #2: ... but their Fig 4 shows the exact opposite, with an increase in Si-Si pair correlations at short distances... so what to believe? this is very confusing and was not addressed despite my comments.

Authors: The authors are not sure what Reviewer #2 means here, but maybe the uncertainty arose from the previous unclear visibility and description of Fig. 2. In contrast to Fig. 2, Fig. 4 shows the results for the so-called "bulk aging" experiments. When a significant time of the totally applied 3 weeks of natural aging was performed in the bulk (1 week) the from the clustering

expected from literature (increase in the Si-Si value) took place. When only 9 minutes of the total 3 weeks of natural aging are in bulk condition a very small signal for Si-Si as for Fig. 2 is observed. We amended the legend to clarify this.

The fact that we see a small signal for Mg after 9 minutes bulk aging (compared to 0 min for Fig. 2) was addressed in the previous revision and is improved in the following:

“The small signals of 9 min bulk aging + 3 weeks nano aging, case (i) in Fig. 4b, may be caused by a slight clustering during the nine minutes of RT preparation at bulk dimension (Mg-Mg). However, large-scale density variations resulting from a methodical artefacts may also cause such minor effects (see Supplementary Fig. 7, Si and Mg, and Supplementary Note 3).”

Reviewer #2: The discussion on the artefact in the APT and associated surface diffusion is really unclear - there are arguments that the increase in Si composition at poles and zone lines is related to chromatic aberrations (see Marquis and Vurpillot, M&M 2008) affecting the Al preferentially, which leads to a local loss of Al and not an increased amount of Si. The paper by Gault et al. mostly demonstrates this for interstitials that can easily jump from one atomic position to another, which is unlikely to be the case between Si and Al, and the mechanism proposed by Oberdorfer et al. relies on the use of potentials that are likely ill-defined for surfaces and on time-scales that may not be realistic. If at play, all the solutes would always be on zone lines! it's not the case. These mechanisms may not be at play at all, and for now, I don't see how the author's argument is convincing or advancing understanding on these fronts. I know this is likely beyond the scope of this paper, but I feel that the authors are using arguments that sound scientific to make a specific point fit their story.

Authors: We agree that the addressed APT artefacts are out of the main scope of the manuscript and it has no importance for the message of the paper. That is why we referred to this minor effect that the studied region of interest can influence slightly our solute distribution measure in the supplementary material. In principal we do not need this in detail, but we intended to show this results as additional information for some critical readers.

For detailed comments see below:

Reviewer #2: The discussion on the artefact in the APT and associated surface diffusion is really unclear - there are arguments that the increase in Si composition at poles and zone lines is related to chromatic aberrations (see Marquis and Vurpillot, M&M 2008) affecting the Al preferentially, which leads to a local loss of Al and not an increased amount of Si.

Authors: In the mentioned paper (Marquis and Vurpillot, M&M 2008) so-called chromatic aberration is described where two species in precipitates are observed spatially de-located systematical from each other in the APT reconstruction due to the from Reviewer #2 mentioned local loss of Al. But we investigate heat treatment states close to solid solutions far from existing precipitates, with different binding environments of the solutes. Therefore the specific effect described in (Marquis and Vurpillot, M&M 2008) does not directly apply, but for sure has the same origin, the preferential retention of high field evaporation solutes. We further follow the explanation in (Gault, 2012, Atom probe Microscopy, 7.4.2 Surface Migration):

“As depicted in Fig. 7.22, zones that display an erroneously high concentration of solute atoms can often be observed in wide-field-of-view atom probe tomography in the vicinity of poles and zone lines, which are regions with high field gradients. One explanation for this increase in Cu or Mg concentration could be a preferential loss of the Al atoms in the vicinity of the pole due to trajectory aberrations. Al may suffer more from such a specific loss, as it is in much higher concentration. Therefore, this apparent increase could be due to Cu and Mg atoms being less affected than matrix atoms by trajectory aberrations occurring at the pole. However, a depletion in Al compared to Cu or Mg would artificially increase the concentration, but not its density.”

The Supplementary Fig. 4, 7 and 8 clearly show increased densities of Si at the (111) pole and the zone lines to it.

Reviewer #2: The paper by Gault et al. mostly demonstrates this for interstitials that can easily jump from one atomic position to another, which is unlikely to be the case between Si and Al, ...

Authors: While it is true that this effect is mostly demonstrated in the mentioned paper by Gault et. al., it is also stated “In conclusion, surface diffusion processes, especially directional walk, was shown to occur for a wide range solute species. This phenomenon primarily affects interstitial elements, but has also been observed for substitutional elements such as Si in steel, impacting the microanalytical capabilities of APT.” Therefore we think this is a possible mechanism to explain the increased density of Si at the (111) pole and the zone lines to it.

Reviewer #2: ... and the mechanism proposed by Oberdorfer et al. relies on the use of potentials that are likely ill-defined for surfaces and on time-scales that may not be realistic. If at play, all the solutes would always be on zone lines! it's not the case.

Authors: For sure not all solutes are at zone lines in our measurements, but the amount should also not be underestimated. We cannot judge the applicability of the used potentials in the paper of Oberdorfer et. al., but comparing the computed stereographic projections with the experimental obtained detector hitmaps give a qualitative match (comparing Oberdorfer et. al. Fig. 7 b to Supplementary Fig. 7 and Oberdorfer et. al. Fig. 7 c to Supplementary Fig. 4 and 8). This lets us consider the proposed mechanism as also reasonable.

Reviewer #2: These mechanisms may not be at play at all, and for now, I don't see how the author's argument is convincing or advancing understanding on these fronts.

Authors: However, somehow the Si atoms are able to move during the experiment by some mechanism. Our point is that this artefact seems to have a lower impact on the spatial analysis for samples with a larger radius as mentioned in the text.

Further we showed with Supplementary Fig. 6 that large scale density variations, likely caused by this artefact, can cause the significant increase of ~20 % for the Si-Si curves in Fig. 2.

Although all this does not affect the message of the paper we intended to show it at minor position in the SM.

Reviewer #3

Reviewer #3: I acknowledge that the authors have responded to each comment. In particular, they have attempted to make discuss the generality of the effect more deeply and have amended the title accordingly, which is appreciated.

Authors: We thank the Reviewer for acknowledging the amendments made.

Reviewer #3: This being said, I still believe that these nice experiments are less broadly relevant than claimed, the influence of non-equilibrium quenched-in vacancies on further kinetics concerning mostly aluminium alloys engineering.

What the authors have highlighted here is that:

- the annihilation rate of non-equilibrium quenched-in vacancies depends on the density of vacancy "sinks" and annihilation sites (which is, of course, not surprising, but these experiments are a nice way to actually measure it).
- these type of in situ (or pseudo in situ) experiments at this scale have limited representativity with respect to what happens in the bulk material. This is similar to what happens with in situ TEM ageing experiments where surface effects are important.

I still believe that presenting these results along those lines would have a much higher impact on the community, rather than trying to write them very generally.

Authors: We thank the reviewer for acknowledging the importance of our finding for Al alloys and we are also sure that our results will have high impact on aluminium alloys engineering. We did our best to increase the visibility of this aspect in a revised version of the manuscript. Although excess vacancy driven diffusion is most often discussed in Al, we do not want to lose the general validity of the mechanisms and its increasing importance in other fields (i.e. aging of Mg alloys). We have intensively discussed the reviewer's argument that the chosen general viewpoint may reduce the visibility in the important Al community. Although Al is addressed throughout the paper, we think that the aluminum relevance is not directly recognizable because of the general title. Thus, beside minor amendments of the abstract and the manuscript we now suggest to directly point to the importance for Al alloys in the title:

“Size-Dependent Diffusion: Material Dimensions Determine Aging in Al-Alloys”

We thank the reviewer for his advice which will help us to increase the potential impact of the paper.

REVIEWERS' COMMENTS:

Reviewer #2 (Remarks to the Author):

I think there will be points on which we can agree to disagree, but the authors have answered my criticisms and concerns, somehow

Reviewer #2:

I think there will be points on which we can agree to disagree, but the authors have answered my criticisms and concerns, somehow.

Authors:

We acknowledge that the reviewer is satisfied with our answers to his/her concerns, even if there are, in some points, different opinions present. We thank the reviewer for the effort to critically review the paper.